# Exploring the Antioxidant Properties of Caffeoylquinic and Feruloylquinic Acids: A Computational Study on Hydroperoxyl Radical Scavenging and Xanthine Oxidase Inhibition

**DOI:** 10.3390/antiox12091669

**Published:** 2023-08-25

**Authors:** Houssem Boulebd, Miguel Carmena-Bargueño, Horacio Pérez-Sánchez

**Affiliations:** 1Department of Chemistry, Faculty of Exact Science, University of Constantine 1, Constantine 25000, Algeria; 2Structural Bioinformatics and High-Performance Computing Research Group (BIO-HPC), Computer Engineering Department, Universidad Católica de Murcia (UCAM), Campus de los Jerónimos 135, 30107 Guadalupe, Spain; mcarmena@ucam.edu (M.C.-B.); hperez@ucam.edu (H.P.-S.)

**Keywords:** caffeoylquinic acid, feruloylquinic acid, hydroperoxyl radical scavengers, xanthine oxidase inhibitors, DFT method, blind docking, molecular dynamics simulations

## Abstract

Caffeoylquinic (5-CQA) and feruloylquinic (5-FQA) acids, found in coffee and other plant sources, are known to exhibit diverse biological activities, including potential antioxidant effects. However, the underlying mechanisms of these phenolic compounds remain elusive. This paper investigates the capacity and mode of action of 5-CQA and 5-FQA as natural antioxidants acting as hydroperoxyl radical scavengers and xanthine oxidase (XO) inhibitors. The hydroperoxyl radical scavenging potential was investigated using thermodynamic and kinetic calculations based on the DFT method, taking into account the influence of physiological conditions. Blind docking and molecular dynamics simulations were used to investigate the inhibition capacity toward the XO enzyme. The results showed that 5-CQA and 5-FQA exhibit potent hydroperoxyl radical scavenging capacity in both polar and lipidic physiological media, with rate constants higher than those of common antioxidants, such as Trolox and BHT. 5-CQA carrying catechol moiety was found to be more potent than 5-FQA in both physiological environments. Furthermore, both compounds show good affinity with the active site of the XO enzyme and form stable complexes. The hydrogen atom transfer (HAT) mechanism was found to be exclusive in lipid media, while both HAT and SET (single electron transfer) mechanisms are possible in water. 5-CQA and 5-FQA may, therefore, be considered potent natural antioxidants with potential health benefits.

## 1. Introduction

Coffee is one of the most popular beverages in the world, and it is known to provide several health benefits [1]. It contains caffeine, which is a natural stimulant that can improve mental alertness, concentration, and mood [2]. Moreover, coffee is rich in antioxidants, such as chlorogenic acid and caffeic acid, which can protect against oxidative stress and inflammation and may lower the risk of various chronic diseases, including type 2 diabetes, liver disease, and some types of cancer [3]. Some studies also suggest that moderate coffee consumption may be associated with a lower risk of cognitive decline, depression, and mortality [4].

5-Caffeoylquinic (5-CQA) and 5-feruloylquinic (5-FQA) acids are two types of phenolic compounds found in abundance in coffee and other plants, which have been shown to possess various biological activities [5,6] (Figure 1). 5-CQA is present in many dicotyledonous plants and is the most abundant phenolic acid in both Arabica and Robusta green coffee beans, accounting for 5–6% of dry beans [7]. 5-FQA, on the other hand, is relatively less abundant in green coffee beans (1%) and is found in many foods, such as black currant, carrot, highbush blueberry, and loquat [8]. Both compounds are derivatives of quinic acid, which is a cyclic carboxylic acid. 5-CQA, also known as chlorogenic acid, has a caffeic acid moiety attached to the quinic acid backbone. 5-FQA has a similar structure but has a ferulic acid moiety attached to the quinic acid backbone. Both compounds contain multiple hydroxyl groups that make them potential antioxidants by donating electrons or hydrogen atoms to free radicals and reducing their reactivity. 5-CQA has potent antioxidant properties and has been shown to protect against oxidative stress and inflammation [9,10]. 5-CQA may also help regulate blood sugar levels and improve insulin sensitivity [11]. 5-FQA has also antioxidant and anti-inflammatory properties [12]. It has been shown to have neuroprotective effects and may help prevent cognitive decline [13,14]. The beneficial effects of these compounds have been the subject of numerous biological studies. A study demonstrated the efficient absorption and rapid metabolism of 5-CQA, as well as some evidence of antioxidant activity in vivo [15]. In addition, the antioxidant properties of six isomers of dicaffeoylquinic acid were analyzed, and the results showed that the isomers had varying antioxidant capacities, with some showing greater activity than others [16]. The molecular mechanisms underlying the antioxidant activities of chlorogenic acid were examined in vitro, and it was found that the compound inhibited the activity of several key transcription factors involved in inflammation and oxidative stress and induced the expression of phase 2 detoxifying enzymes, which are important for protecting cells against oxidative damage [17]. In addition, a study showed that a rosmarinic acid derivative has potent antioxidant activity that may have applications in the management of diabetes and hypertension [18]. Despite the numerous investigations conducted, the precise mechanisms by which 5-CQA and 5-FQA exert their antioxidant activity, especially in their direct reactions with free radicals, remain unclear. Furthermore, there has been no comparison of the antioxidant potentials of these two compounds. Therefore, further studies are needed to fully understand the antioxidant capabilities of these biologically active molecules.

Xanthine oxidase (XO) is an enzyme found in the liver and other tissues that plays a crucial role in purine metabolism. It catalyzes the oxidation of hypoxanthine to xanthine and xanthine to uric acid, generating reactive oxygen species (ROS) as by-products. High levels of XO activity and ROS production have been associated with oxidative stress, inflammation, and various diseases, including cardiovascular disease and diabetes [19]. Several components of coffee, including caffeine and chlorogenic acid, have been shown to inhibit XO activity and reduce ROS production [20,21]. The XO inhibition potential of 5-CQA, in particular, has been the subject of several experimental studies; however, to our knowledge, 5-FQA has not yet been considered. For example, Arshad et al. investigated the XO inhibitory activity of eight phenolic compounds, including 5-CQA, using in vitro and computational methods, which revealed that these compounds possess varying levels of XO inhibitory activity, potentially lowering uric acid levels and combating hyperuricemia [21]. On the other hand, Wan et al. explored the mechanisms by which eight caffeoylquinic acids (CQAs) inhibit XO in vitro, showing that diCQAs have higher inhibitory activity than monoCQAs due to hydrophobic interaction and hydrogen bonding and that CQAs preferentially bind to the flavin adenine dinucleotide region of XO [22]. In another work, Mohamed Isa et al. studied the XO inhibitory activity of the methanol extract of Plumeria rubra Linn flowers, which contain large amounts of phenol and flavonoids, showing that the extract exhibits strong XO inhibitory activity in vitro and can reduce serum uric acid levels in rats without toxicity. Despite these numerous studies on the potential inhibition of XO by 5-CQA, the exact mechanism of the interaction between this molecule and the enzyme is still unclear. Furthermore, given the structural similarity between 5-CQA and 5-FQA, it is reasonable to assume that 5-FQA also has potent XO inhibitory activity. Therefore, further research is needed to determine the XO inhibitory potential of 5-CQA and 5-FQA.

In this context and in order to explore in depth the antioxidant potential of 5-CQA and 5-FQA as well as to compare their capacities, this study focused on the evaluation of the reactivity toward the hydroperoxyl radical (HOO^•^) in physiological environments and on the mechanism of XO inhibition. The HOO^•^ radical is a moderately reactive ROS with a half-life of about a few seconds in biological cells, and it is the simplest of the peroxyl radicals that can cause significant damage to cells [23,24]. Due to these characteristics, it is a suitable target for examining antiradical mechanisms [25]. In this study, the main antiradical mechanisms were examined taking into account the influence of pH and physiological environments at all possible sites of 5-CQA and 5-FQA. All the work was carried out using DFT calculations at the M06-2X/6-311++G(d,p) level, which is one of the most accurate methods for describing radical reactions in solution. XO inhibition, on the other hand, was examined by blind docking and molecular dynamics simulations. These in silico approaches are useful methods for studying the interactions between a ligand and a target protein and can provide valuable information on the binding mechanism and stability of the complex. The findings of this study show that 5-CQA and 5-FQA are potent radical scavengers, with 5-CQA being more potent than 5-FQA. Both compounds were found to be more potent than the common antioxidants Trolox and BHT under physiological conditions. Furthermore, both compounds inhibit XO enzymes in the same way as quercetin.

## 2. Computational Details

### 2.1. Quantum Chemistry Calculations

Phenolic compounds exhibit their antiradical action through three distinct processes: HAT (hydrogen atom transfer), RAF (radical adduct formation), and SET (single electron transfer) [26,27,28,29,30]. HAT involves the transfer of a hydrogen atom from the antioxidant to the free radical, and it can occur in both polar and lipid environments. The Gibbs free energy of the reaction describes this one-step process. On the other hand, RAF is a one-step process where free radicals combine with antioxidants, forming the product [antioxidant-free radical]^•^ [31]. The Gibbs free energy of the reaction describes this process. Unlike HAT and RAF, SET involves an electron transfer and only occurs in polar environments [32]. The phenolic OH bonds of the antioxidant can exist in both protonated and deprotonated forms in these environments, enabling electron transfer from the neutral or anionic state of the antioxidant. However, SET from an undissociated phenol is often not feasible, making it significant only for deprotonated phenols. The Gibbs free energy of electron transfer describes the dominant step in this pathway. These mechanisms are represented by the following equations:$$Ar-OH+R^{∙}\to {Ar-O}^{∙}+ROH\;\left(\mathrm{HAT}\right)$$
$$Ar-OH+R^{∙}\to {\left[Ar-OH-R\right]}^{∙}\;\left(\mathrm{RAF}\right)$$
$$Ar-OH\to {Ar-O}^{-}+H^{+};{Ar-O}^{-}+R^{∙}\to {Ar-O}^{∙}+R^{-}\;\left(\mathrm{SET}\right)$$

In this work, density functional theory (DFT) computations were carried out using the Gaussian 09 version E.01 software [33]. The M06-2X functional, which is highly parametrized and includes empirical exchange-correlation, was used in combination with the 6-311++G(d,p) basis set, as it is known to be one of the most accurate approaches for calculating rate constants for radical reactions in solution [34,35,36]. To account for solvation effects in physiological media, the SMD model was employed [37]. Proton affinity (PA) values of OH groups were computed as described in the literature [38]. The pKa values have been calculated according to the literature using the following equation [39]:$$AH+{OH}^{-}\left(3H_{2}O\right)\to A^{-}\left(H_{2}O\right)+3H_{2}O$$
$${pK}_{a}=\frac{{∆G}_{sol}}{RTln\left(10\right)}+14+3\log\left[H_{2}O\right]$$
where ∆G_sol_ is the Gibbs free energy of the reaction in solution, R is the gas constant, and T is the temperature (298.15 K).

The quantum mechanics-based test for overall free radical scavenging activity (QM-ORSA) methodology was employed for the kinetic study, which was validated experimentally [25,40,41]. The rate constants (k) were calculated at 298.15 K using conventional transition state theory (TST) [42,43,44,45,46].
$$k=\sigma \kappa \frac{k_{B}T}{h}e^{\frac{-{\Delta G}^{\neq }}{RT}}$$
where σ, κ, kB, h, ∆G^≠^ are reaction symmetry numbers, tunneling corrections, Boltzmann constant, Planck constant, and Gibbs free energy of activation, respectively [47,48,49]. The tunneling correction that refers to the ratio between quantum mechanics and classical mechanical rates of barrier crossing was calculated using the Eckart barrier [47].

The ratio of the probability of a specific pathway to the total probability of all possible pathways, branching ratio (Γ), was determined using the equations below:$$\Gamma_{i}=\frac{k_{i}}{k_{overall}}\times 100$$
where k_i_ and k_overall_ are the rate constants of a specific reaction path and the sum of the rate constants of all reaction paths, respectively.

The Gibbs free energy of activation (${∆G}_{SET}^{\neq }$) for the single electron transfer (SET) mechanism was predicted using the Marcus theory approach [50], with λ approximated as $\lambda \approx {\Delta E}_{SET}+{∆G}_{SET}^{0}$.
$${∆G}_{SET}^{\neq }=\frac{\lambda }{4}{\left(1+\frac{{∆G}_{SET}^{0}}{\lambda }\right)}^{2}$$
where ${∆G}_{SET}^{0}$ is the Gibbs free energy of the reaction.

Finally, the apparent rate constants (k_app_) were corrected using the Collins‒Kimball theory to account for diffusion limitation [51].

### 2.2. Blind Docking Consensus Procedure

We applied the blind docking (BD) consensus approach to study the interactions between both ligands and XO. BD is a method that scans the whole protein surface finding potential hotspots for a specific ligand [52]. First, we obtained the crystallized structure of the bovine xanthine oxidase described in the Protein Data Bank (PDB), whose code is 3nvy (https://www.rcsb.org/structure/3nvy accessed on 15 September 2021). The PDB file obtained was processed using Maestro tools: Protein Preparation Wizard and System Builder. The ligand was processed with LigPrep, the tool of Maestro designed to prepare small molecules. The format of the structure obtained was mol2. All the mol2 files generated by Maestro used the force field OPLS3e [53]. Protein and ligands were processed with ADT to obtain a pdbqt file. The force field of the pdbqt file was Gasteiger.

BD calculations were performed using AutoDock Vina [54] and Lead Finder [55]. The grid box size of all runs had measurements of 30 × 30 × 30 Angstroms as the default parameter. All the runs, including the BD consensus, were performed using an alpha version 1.0 of metascreener software (https://github.com/bio-hpc/metascreener) The scoring function of AD and LF considered the following parameters: Lennard-Jones interactions term (LJ), hydrogen bonds (H-bonds), electrostatic interactions, hydrophobic stabilization, entropic penalty due to the number of rotatable bonds, and internal energy of the ligand. Maestro software (Maestro-Desmond Interoperability Tools, Schrödinger, New York, NY, USA, 2020) was also used to calculate the interactions between ligand and protein residues.

### 2.3. Molecular Docking Simulations

The most relevant pose of each ligand obtained in the BD was selected to run molecular dynamic simulations (MD). Three MDs were carried out using Maestro-Desmond software version 2020-4 (Desmond Molecular Dynamics System, D. E. Shaw Research, New York, NY, 2020. Maestro-Desmond Interoperability Tools, Schrödinger, New York, NY, USA, 2020). Both complexes created were immersed in a box filled with water molecules using the simple point charge (SPC) scheme. The measurements of the box were 10 × 10 × 10 Å. Sodium ions were added to neutralize charges. Ions of chlorine and sodium were added to obtain a final NaCl concentration of 0.150 M. Periodic boundary conditions were used, and a cutoff of 9 Å was established for van der Waals interactions. The particle mesh Ewald (PWE) method with a tolerance of 10-9 was used in the electrostatic part. The energy minimization was carried out by 2000 steps using the steepest descent method with a threshold of 1.0 kcal/mol/Å. The NPT simulations were realized at 300 K with the Nose-Hoover algorithm [56], and the pressure was maintained at 1 bar with the Martyna‒Tobias‒Klein barostat [57]. The OPLS3e force field was used in all the MDs and the duration of both MDs was 100 ns.

## 3. Results and Discussion

### 3.1. Evaluation of Radical Scavenging Mechanisms

#### 3.1.1. Acid-Base Equilibrium at Physiological pH

Understanding the behavior of phenolic groups at physiological pH is critical in studying free radical scavenging properties in water. While the pKa values of 5-CQA are available in the literature [58], the same cannot be said for 5-FQA. To address this, we followed a procedure outlined in a published paper to calculate the pKa of 5-FQA [39]. We included the obtained pKa values and PA values of the OH groups of both compounds in Figure 2. Our findings indicate that for both 5-CQA and 5-FQA, the COOH group is the preferred deprotonation site, followed by the phenolic OH group at position 13, which has the lowest PA value. In water, the mono-anionic form dominates for both compounds, with a non-negligible amount of the dianionic form present. The neutral form is absent under these conditions, and therefore, we only considered the mono-anionic and dianionic forms in our study. However, in the lipid environment, only the neutral form exists, and we considered only this form in our thermodynamic and kinetic studies.

#### 3.1.2. Thermodynamic Evaluation of the Antiradical Mechanisms

Figure 3 shows the ΔG values of the reaction between 5-CQA and 5-FQA with the HOO^•^ radical in both aqueous and lipid media. In water, the HAT reaction of the non-aromatic OH groups was found to be energetically unfavorable, as indicated by positive ΔG values (31.9–45.1 kcal/mol). This suggests that these OH groups are not effective for antiradical action. Similarly, the RAF reaction at both C8 and C9 positions also showed positive ΔG values, making this mechanism less likely to occur. However, the RAF reaction at the C8 position was almost isergonic (2.1–2.7 kcal/mol), indicating that this process can take place. As a result, the kinetic study also considered the RAF mechanism at C8.

On the other hand, the aromatic OH groups had negative ΔG values for both 5-CQA and 5-FQA (−5.8 to −13.0 kcal/mol), suggesting that they are the most likely sites for the HAT process. In terms of the SET mechanism, both forms showed positive ΔG values. However, the dianionic form exhibited the lowest ΔG values (2.0–4.8 kcal/mol vs. 29.3–29.8 kcal/mol), indicating that this mechanism could also be possible for this form.

When considering pentyl ethanoate, where only the neutral form exists, the HAT reaction was found to be thermodynamically favorable for the phenolic OH groups (−1.4 to −6.9 kcal/mol). In contrast, the RAF reaction was an endogenous process for both compounds (4.4 and 4.8 kcal/mol). Despite this, the kinetic study also examined the RAF reaction at the C8 position.

Overall, the results indicate that the HAT mechanism is more likely to occur in the aromatic OH groups, while the RAF mechanism is less favorable. However, the RAF reaction at the C8 position is also possible. The SET mechanism was found to be less likely but could still be active for the dianionic form.

#### 3.1.3. Reaction Kinetics under Physiological Conditions

##### Effects of Polar Physiological Media

Using the QM-ORSA protocol [25], and based on the thermodynamic evaluation, we calculated the overall and individual rate constants for the reaction between 5-CQA and 5-FQA with the HOO^•^ radical in water at physiological pH. The results are presented in Table 1, and the localized transition states can be seen in Figure 4. We found that both 5-CQA and 5-FQA reacted with high-rate constants of 2.68 × 10^8^ M^−1^s^−1^ and 2.28 × 10^7^ M^−1^s^−1^, respectively. However, the reaction of 5-CQA is about 10 times faster than that of 5-FQA. Analysis of the data in Table 1 reveals that the SET mechanism is exclusive in the case of 5-FQA (Γ = 100%), while the HAT mechanism at 12OH plays a dominant role in the case of 5-CQA (Γ = 70%), despite the high-rate constant of the electron transfer process (7.95 × 10^7^ M^−1^s^−1^). This suggests that the deprotonated catechol moiety in water is highly reactive toward radical species. Similar findings were also observed for other catechol derivatives, such as 5-O-methylnorbergenin [59], quercetins [60], caftaric acid [61], and anthocyanidins [62].

Moreover, it is worth noting that the overall rate constants of 5-CQA and 5-FQA are comparable to that of the reference antioxidant ascorbic acid (k = 9.97 × 10^7^ M^−1^ s^−1^) [25] and the structurally similar system caftaric acid (k = 9.09 × 10^8^ M^−1^ s^−1^) [61] but higher than that of Trolox (1.13 × 10^5^ M^−1^s^−1^) [63], BHT (2.51 × 10^5^ M^−1^s^−1^) [64], carnosic acid (k = 4.73 × 10^6^ M^−1^s^−1^) [65], and cannabidiol (k = 9.09 × 10^6^ M^−1^s^−1^) [66]. This indicates that 5-CQA and 5-FQA are potent antiradical agents under polar physiological conditions.

##### Effects of Lipid-Like Physiological Media

Since the SET process is not feasible in nonpolar media, it was not considered in the lipid environment, thus the study focused only on the RAF and HAT mechanisms. The results obtained from the reaction of 5-CQA and 5-FQA with the HOO^•^ radical in pentyl ethanoate are presented in Table 2, and the localized TSs are shown in Figure 5. As we can see, RAF at the C8 position occurs with a low-rate constant that does not contribute to the overall rate constant for the two compounds studied (Γ = 0%). The HAT mechanism, on the other hand, is dominant under these conditions. The overall rate constants of 5-CQA and 5-FQA are 2.09 × 10^6^ and 4.10 × 10^4^ M^−1^s^−1^, respectively, reflecting the potent free radical scavenging activity of these compounds compared with typical antioxidants, such as Trolox (1.00 × 10^5^ M^−1^s^−1^) [63] and BHT (1.70 × 10^4^ M^−1^s^−1^) [64]. These values are also higher than values for recognized antioxidants such as caftaric acid (k = 1.82 × 10^3^ M^−1^ s^−1^) [61], carnosic acid (k = 5.70 × 10^3^ M^−1^s^−1^) [65], and cannabidiol (k = 2.60 × 10^3^ M^−1^s^−1^) [66], indicating that 5-CQA and 5-FQA are potent antioxidants in lipid media. As observed in water, the reactivity of 5-CQA is higher than that of 5-FQA, indicating that the former is a better antioxidant than the latter in both polar and lipid media.

In summary, the findings revealed that both 5-CQA and 5-FQA react with the HOO^•^ radical in physiological media at high-rate constants, with 5-CQA reacting faster than 5-FQA. Mechanistically, 5-CQA reacts primarily via the HAT mechanism in polar and lipidic media, whereas 5-FQA appears to be more active in water via the SET mechanism rather than the HAT mechanism. The overall rate constants of both compounds are comparable to or greater than those of reference antioxidants, suggesting that these compounds are promising candidates for further study as potential antiradical agents in physiological media.

### 3.2. Blind Docking and Molecular Dynamics Simulations

The results of the BD calculation (Figure 6) show how the first three hotspots have a similar docking score (5-CQA: −11.21 kcal/mol to −9.94 kcal/mol, 5-FQA: −10.12 kcal/mol to −9.47 kcal/mol). The first hotspot is the enzyme’s active site, whose substrate is quercetin [67]. Figure 7 shows the residues that interacted with both ligands. Regarding the interactions with residues, some of them are common with the quercetin complex: E1261 interacted with both ligands and Glu802, Leu873, Arg880, and Phe914. Phe1009, Thr1010, Val1011, and Leu1014 interacted only with 5-CQA. The average docking score of each compound was as follows: −11.21 kcal/mol for the 5-CQA and −10.12 kcal/mol for the 5-FQA.

The results of the MD show how the protein reached a stable conformation in both complexes. The maximum RMSD value of the protein was 2.7 Å in both complexes. Regarding the MD with the protein without ligand, the maximum RMSD value was 1.8 Å. Moreover, the ligand stayed in a stable pose in the active site: 5-CQA had a maximum RMSD value of 4.5 Å, and 5-FQA had a value of 3.6 Å (Figure 8).

The main interactions of the 5-CQA along the trajectory were as follows: ionic interactions—Glu1261; hydrophobic interactions—Phe914, Phe1009, Val1011, and Ala1078; and polar interactions—Gln767, Thr1010, Gln1040, Ala1078, Ser1080, and Ser1082 (Figure 9). The main interactions of the 5-FQA along the trajectory were as follows: ionic interactions—Arg912, Lys1045, Asp1191, and Glu1261; polar interactions—Thr1077, Ser1080, and Ser1082; and water interactions—Gly913 (Figure 10). These results show the high stability of 5-CQA and 5-FQA and Xanthine Oxidase complexes, and we could infer that both ligands could be inhibitors of the enzyme.

## 4. Conclusions

The free radical scavenging activity and mechanism as well as the XO inhibition potential of 5-CQA and 5-FQA were fully investigated. The radical scavenging activity was assessed by modeling the reactivity of 5-CQA and 5-FQA toward the HOO^•^ radical following the main antiradical mechanism under physiological conditions. The three mechanisms—HAT, RAF, and SET—were considered for all possible positions of the molecules, taking into account the influence of physiological pH. The results revealed that 5-CQA and 5-FQA were potent antioxidants with a greater capacity to scavenge hydroperoxyl radicals in polar and lipidic media than common antioxidants, such as Trolox and BHT. 5-CQA with the catechol moiety was found to be more active than 5-FQA. Furthermore, 5-CQA and 5-FQA exist mainly in dissociated form at physiological pH, which is important for their antioxidant activity. It was also revealed that the mechanism of 5-CQA action was mainly the HAT in both physiological environments, while the mechanism of 5-FQA was environment-dependent, with the SET mechanism becoming more important in polar environments.

On the other hand, the inhibition potential of XO was evaluated using BD and MD simulations. The BD results showed that the active site of the enzyme was the first hotspot with the best docking score for 5-CQA and 5-FQA. The 100 ns MD simulation confirmed the stability of the complexes and revealed that both compounds interact mainly with the same residues as quercetin.

These results suggest that 5-CQA and 5-FQA are potent free radical scavengers and XO inhibitors and may have potential applications as natural antioxidants in various fields, including the food and medicinal industries.

## Figures and Tables

**Figure 1 antioxidants-12-01669-f001:**
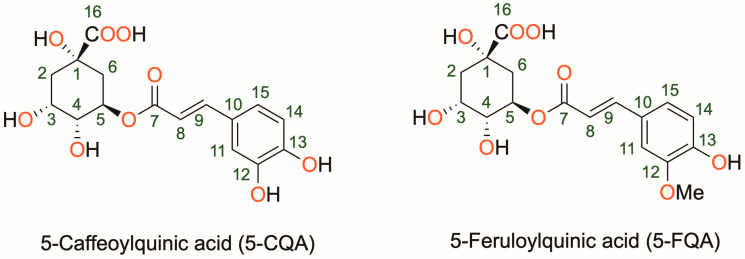
Molecular structure of the investigated phenolic acid derivatives (5-CQA = 5-caffeoylquinic acid and 5-FQA = 5-feruloylquinic acid).

**Figure 2 antioxidants-12-01669-f002:**
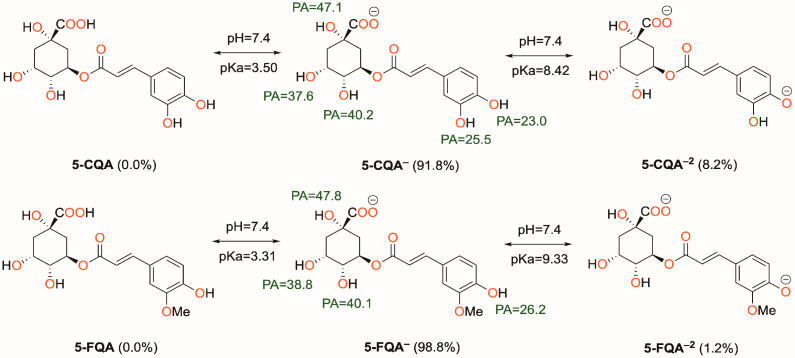
Acid-base equilibrium of 5-CQA and 5-FQA at physiological pH (7.4). PA = proton affinity.

**Figure 3 antioxidants-12-01669-f003:**
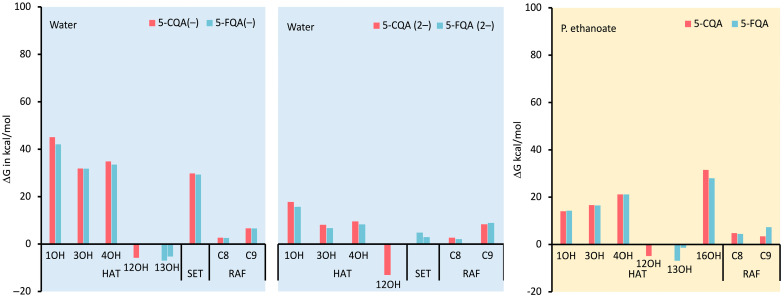
ΔG values calculated in kcal/mol for the reactions of 5-CQA and 5-FQA and their dissociated forms with the HOO^•^ radical according to possible antiradical mechanisms under physiological conditions (water and pentyl ethanoate). HAT = hydrogen atom transfer, SET = single electron transfer, and RAF = radical adduct formation.

**Figure 4 antioxidants-12-01669-f004:**
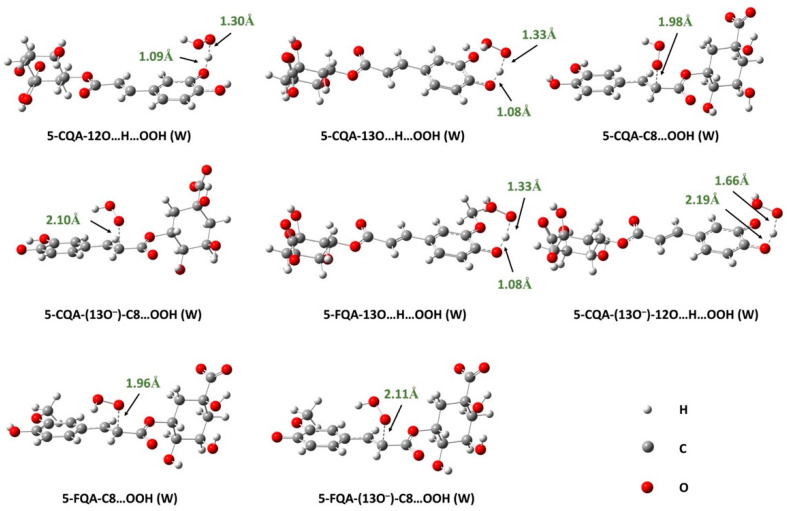
Optimized TS structures of the reaction of 5-CQA and 5-FQA with HOO^•^ radical following HAT (hydrogen atom transfer) and RAF (radical adduct formation) mechanisms in water (W).

**Figure 5 antioxidants-12-01669-f005:**
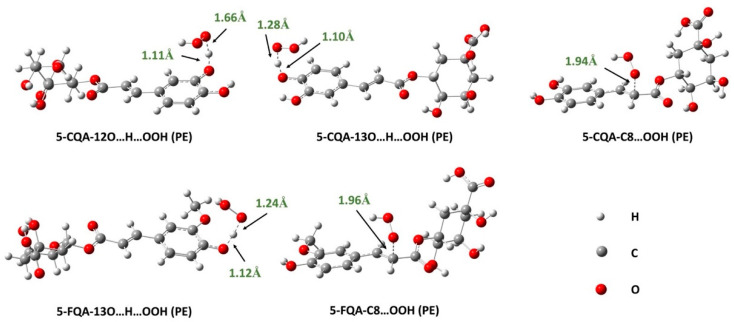
Optimized TSs structures of the reaction of 5-CQA and 5-FQA with HOO^•^ radical following HAT (hydrogen atom transfer) and RAF (radical adduct formation) mechanisms in pentyl ethanoate (PE).

**Figure 6 antioxidants-12-01669-f006:**
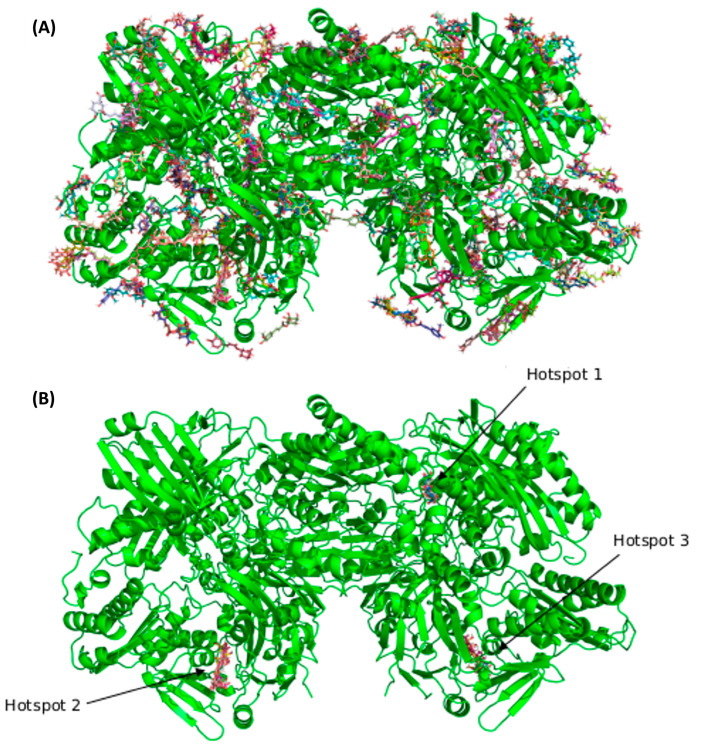
(**A**) Blinding the docking complex with all hotspots from the BD consensus. In (**B**), only the best three hotspots are represented. Illustrative images were prepared using the Pymol software version 2.5.0.

**Figure 7 antioxidants-12-01669-f007:**
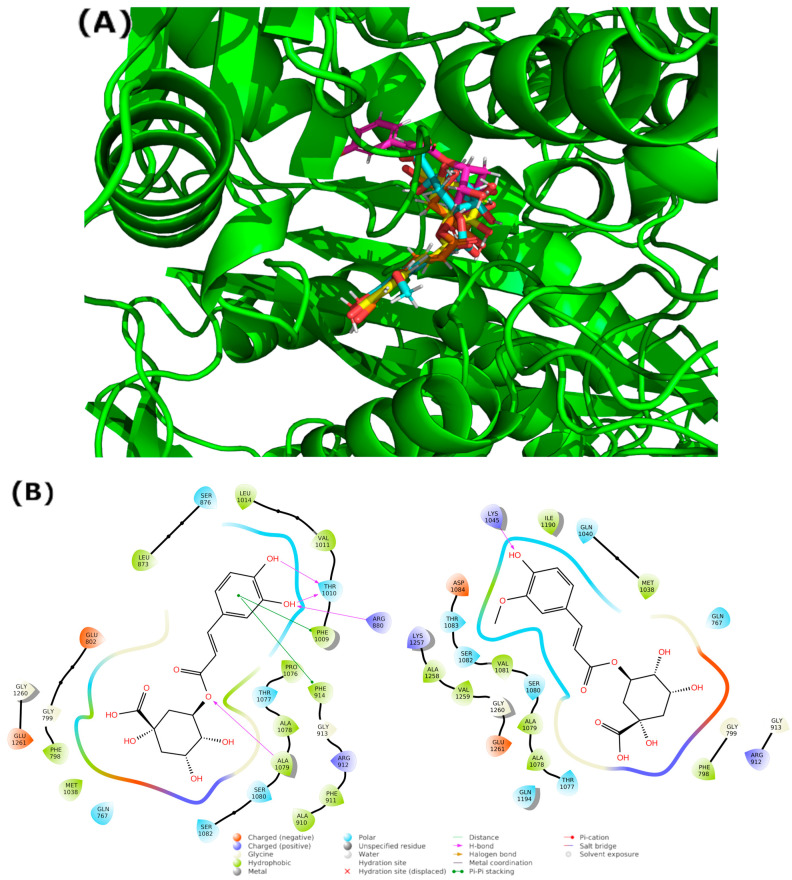
(**A**) Blinding the docking complex with hotspot number 1 resulted from the BD consensus. Illustrative images were prepared using the Pymol software version 2.5.0. (**B**) Two-dimensional protein‒ligand interaction diagram generated between xanthine oxidase and 5-CQA (left) and 5-FQA (right), using the Ligand Interaction script in Maestro (Schrödinger, Inc., www.schrodinger.com accessed on 20 June 2022).

**Figure 8 antioxidants-12-01669-f008:**
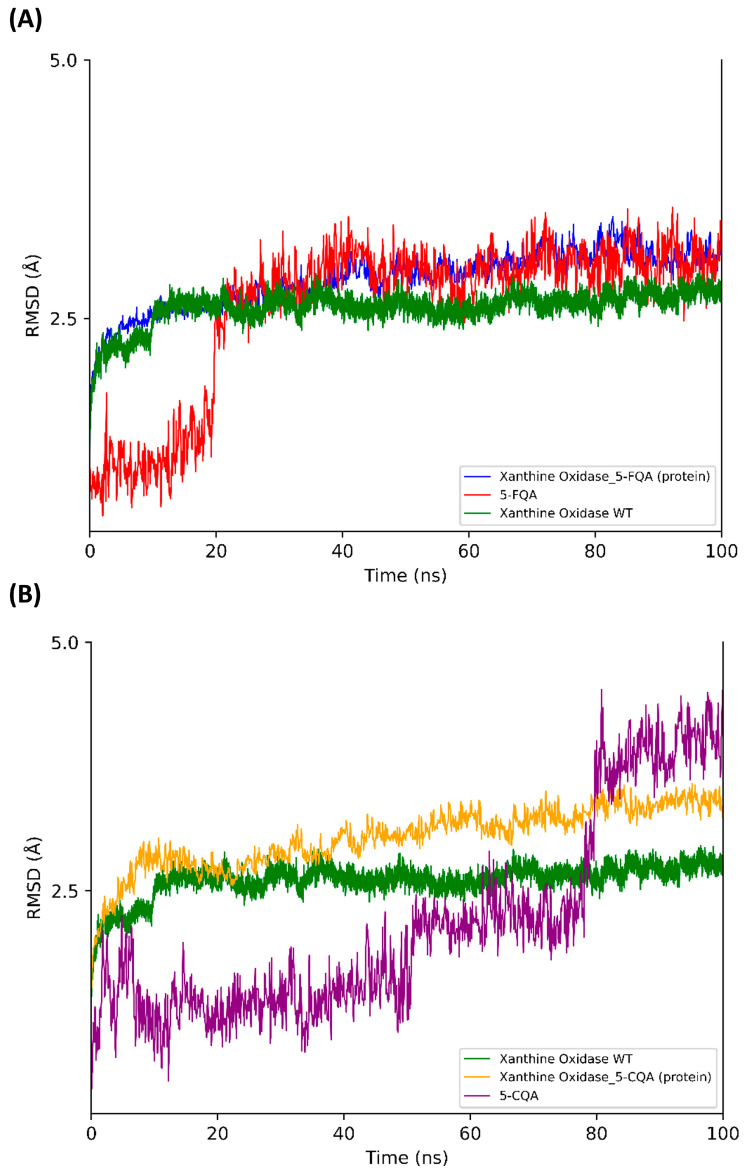
The root-mean-square deviation (RMSD) obtained for xanthine oxidase (blue (**A**) and orange (**B**)) bound to 5-FQA (red) or 5-CQA (purple) plotted with respect to the initial pose. The RMSD of the xanthine oxidase WT is represented in both plots (**A**,**B**) in green.

**Figure 9 antioxidants-12-01669-f009:**
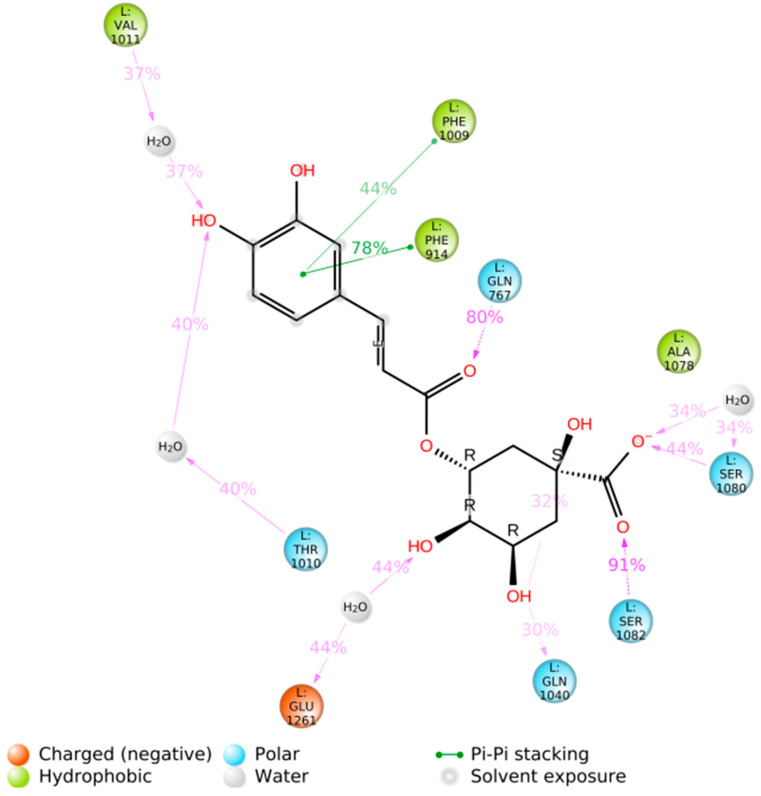
Two-dimensional protein‒ligand interaction diagram generated between xanthine oxidase and 5-CQA, using the Simulation Interactions Diagram script in Maestro (Schrödinger Inc., www.schrodinger.com accessed on 20 June 2022).

**Figure 10 antioxidants-12-01669-f010:**
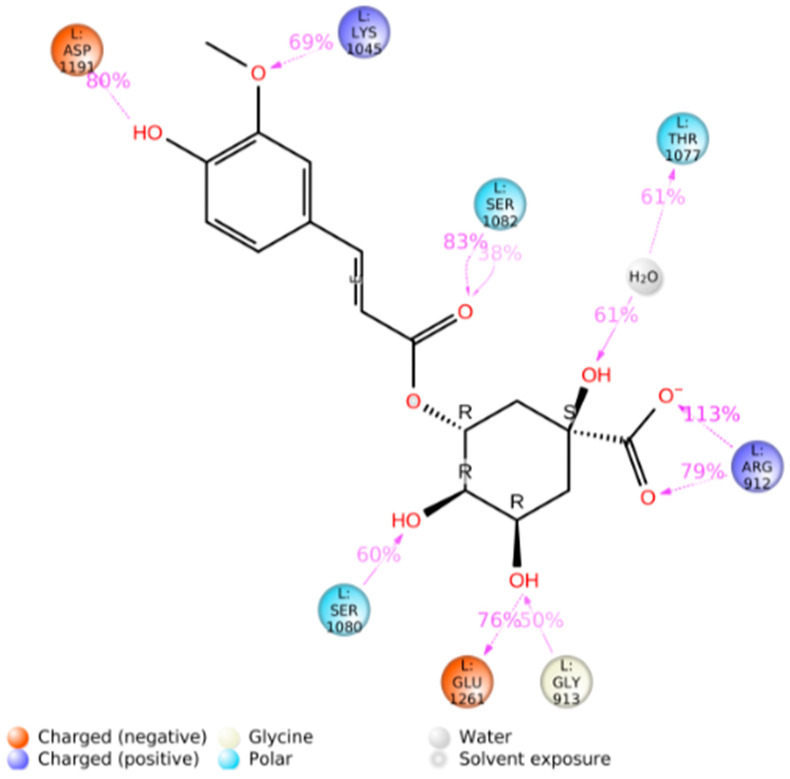
Two-dimensional protein‒ligand interaction diagram generated between xanthine oxidase and 5-FQA, using the Simulation Interactions Diagram script in Maestro (Schrödinger Inc., www.schrodinger.com accessed on 20 June 2022).

**Table 1 antioxidants-12-01669-t001:** Calculated Gibbs free energy of activation (ΔG^≠^ in kcal/mol), tunneling correction (κ), branching ratio (Γ in %), and rate constants (*k* in M^−1^s^−1^) of the reaction of 5-CQA and 5-FQA with HOO^•^ radical following the thermodynamically favorable antiradical mechanisms in water at physiological pH.

Comp.	Mechanisms		State	ΔG^≠^	κ	k_app_	f ^a^	k_f_ ^b^	Γ	k_overall_
5-CQA	HAT	12OH	5-CQA^–^	16.8	205.7	6.60 × 10^2^	0.918	6.06 × 10^2^	0	2.68 × 10^8^
		13OH		17.5	436.8	8.40 × 10^1^		7.71 × 10^1^	0	
	RAF	C8		19.6	1.4	2.50 × 10^−2^		2.30 × 10^−2^	0	
	HAT	12OH	5-CQA^–2^	2.0	1.2	2.30 × 10^9^	0.082	1.89 × 10^8^	70	
	RAF	C8		14.2	1.0	2.40 × 10^2^		1.97 × 10^1^	0	
	SET			4.8	4.0 ^c^	9.70 × 10^8^		7.95 × 10^7^	30	
5-FQA	HAT	13OH	5-FQA^–^	17.4	838.8	1.70 × 10^2^	0.988	1.68 × 10^2^	0	2.28 × 10^7^
	RAF	C8		17.5	1.4	1.20 × 10^0^		1.19 × 10^0^	0	
	RAF	C8	5-FQA^–2^	10.8	1.0	7.80 × 10^4^	0.012	9.36 × 10^2^	0	
	SET			3.7	1.1 ^c^	1.90 × 10^9^		2.28 × 10^7^	100	

^a^ mole fraction; ^b^ k_f_ = f.k_app_; ^c^ nuclear reorganization energy (λ).

**Table 2 antioxidants-12-01669-t002:** Calculated Gibbs free energy of activation (ΔG^≠^ in kcal/mol), tunneling correction (κ), branching ratio (Γ in %), and rate constants (*k* in M^−1^s^−1^) of the reaction of 5-CQA and 5-FQA with HOO^•^ radical following the thermodynamically favorable antiradical mechanisms in pentyl ethanoate.

Comp.	Mechanisms		ΔG^≠^	κ	k_app_	Γ	k_overall_
5-CQA	HAT	12OH	8.7	0.7	1.80 × 10^6^	86	2.09 × 10^6^
		13OH	5.7	0.0	2.90 × 10^5^	14	
	RAF	C8	19.7	1.5	3.40 × 10^−2^	0	
5-FQA	HAT	13OH	7.9	0.0	4.10 × 10^4^	100	4.10 × 10^4^
	RAF	C8	16.0	1.5	1.60 × 10^1^	0	

## Data Availability

Data is contained within the article or Appendix A.

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
