# Peer review of "Exploring the Antioxidant Properties of Caffeoylquinic and Feruloylquinic Acids: A Computational Study on Hydroperoxyl Radical Scavenging and Xanthine Oxidase Inhibition"

_antioxidants, 2023, doi:10.3390/antiox12091669_

Round 1

Reviewer 1 Report

Houssem et. al paper presents a theoretical study involving two polyphenolic compounds on two main directions - hydroperoxyl scavenging and xanthine oxidase inhibition.

The paper is a typical one in the field of antioxidants linked with the in silico calculations. 

I have no observations on the quantum calculations. 

In the docking section I would recommend to write the amino acids using the three letter code. I find it easier to read and I consider it more common. Why blind docking and not directly on the active site? Please give a short motivation.

Regarding the molecular dynamics I have some observations:

-Figure 8: divide it in to two figures  (plot only a pair on the same figure) = XO+5FQA vs 5FQA and in the next figure XO+5CQA vs. 5CQA. Maybe using a thinner line (if the plotting software can do this).

Author Response

Point 1: Houssem et. al paper presents a theoretical study involving two polyphenolic compounds on two main directions - hydroperoxyl scavenging and xanthine oxidase inhibition.
The paper is a typical one in the field of antioxidants linked with the in silico calculations. 
I have no observations on the quantum calculations. 
In the docking section I would recommend to write the amino acids using the three letter code. I find it easier to read and I consider it more common. Why blind docking and not directly on the active site? Please give a short motivation.

Answer 1:  We appreciate the referee's thoughtful comments and valuable input. In response to the suggestion of writing amino acids using the three-letter code, we concur that this notation can enhance readability and is indeed more common in the scientific community. Future versions of our work will incorporate this change.
Addressing the question about our choice of blind docking over directly focusing on the active site, we would like to clarify that blind docking provides a more comprehensive view of potential binding pockets within a protein. This technique allows us to investigate not only the active site but also potential allosteric pockets where the ligand might bind. The rationale behind our decision to perform blind docking rather than focused docking is to identify the pockets—both binding and allosteric sites—with the highest probability of ligand binding. By doing so, we can develop a broader understanding of potential ligand-protein interactions, beyond just the active site. We hope this clarifies our approach and we thank the referee for encouraging this elaboration.

Point 2: Regarding the molecular dynamics I have some observations:
-Figure 8: divide it in to two figures  (plot only a pair on the same figure) = XO+5FQA vs 5FQA and in the next figure XO+5CQA vs. 5CQA. Maybe using a thinner line (if the plotting software can do this).

Answer 2: We appreciate the reviewer's suggestions regarding the presentation of Figure 8. We agree that splitting the figure into two distinct plots—Plot A representing XO+5FQA vs. 5FQA and Plot B illustrating XO+5CQA vs. 5CQA—will aid in clarity and focus. Additionally, we acknowledge the recommendation to employ thinner lines in these plots to enhance visibility and reduce potential clutter. The suggested modifications have been implemented in the revised version of our manuscript. Thank you for guiding us to improve the visual representation of our data.

Reviewer 2 Report

Moderate English editing required

Author Response

By the present paper the authors aim at investigating the capacity and mode of action of Caffeoylquinic (5-CQA) and Feruloylquinic (5-FQA) acids, phenolic compounds which, as they affirm, are abundant in coffee, and are known to exhibit diverse biological activities, in particular potential antioxidant effects. They are derivatives of quinic acid, a cyclic carboxylic acid, exhibiting similar structures: 5-CQA, shows a caffeic acid moiety attached to the quinic acid backbone, while 5-FQA presents a ferulic acid moiety attached to the quinic acid backbone. 5-CQA and 5-FQA are natural antioxidants acting as hydroperoxyl radical scavengers and xanthine oxidase (XO) inhibitors.
Starting from these bases, the authors conducted an interesting and wide study investigating:
1.The hydroperoxyl radical scavenging potential using thermodynamic and kinetic calculations. 
2. the inhibition capacity toward XO enzyme by Blind docking and molecular dynamics simulations 
and reported interesting results as follows:
1. 5-CQA and 5-FQA exhibit potent hydroperoxyl radical scavenging capacity in both polar and lipidic physiological media, with rate constants higher than those of common antioxidants such as Trolox and BHT. 
2. 5-CQA carrying catechol moiety significantly more potent than 5-FQA in both physiological environments. 
3. Both compounds showed good affinity with the active site of the XO enzyme forming stable complexes. 
However, the study, in this reviewer’s opinion, exhaustively conducted, as already affirmed, by a theoretical point of view, completely lacks experimental basis from all point of views. Thus the authors are asked to add and clarify the following points:
1. Which the novelty and the limitation of their study, this latter better in a separate paragraph, and the purpose for the future. 

Answer 1: We thank the reviewers for their valuable comments. Our study focuses on the antioxidant activity and XO inhibition mechanisms of 5-CQA and 5-FQA, using state-of-the-art in silico tools. This research is the first report on these compounds and highlights their potential therapeutic applications. In addition, we have undertaken a comprehensive comparative analysis of the activities exhibited by 5-CQA and 5-FQA, which to our knowledge has not been explored in the literature. This comparative study aims to provide more in-depth information on the distinct properties and potential benefits of these two compounds, thereby contributing to the growing body of knowledge in this area.

2. The authors report that these compounds are present in coffee but, as is well known, coffee is a generic term underlying a lot of variations and qualities. The authors must specify the type of coffee where these compounds are present and most of all their concentration.

Answer 2: We thank the reviewer for this constructive comment. The natural presence of 5-caffeoylquinic acid (5-CQA) and 5-feruloylquinic acid (5-FQA) was briefly described in the introduction.

3.    The authors affirm throughout the text that the experiments were conducted under physiological condition. However, they never specify these conditions and in particular the physiological pH that as well known is different in plants with respect to humans.

Answer 3: We thank the reviewer for this valuable comment. The computational work was performed according to an established protocol developed by Galano et al (J. Comput. Chem. 2013, 34, 2430-2445). In this protocol, physiological conditions are be mimicked using water at pH 7.4 and the solvent pentyl ethanoate for polar and lipid media, respectively. Further details of the protocol can be found in Galano's original work (J. Comput. Chem. 2013, 34, 2430-2445).

4.    Finally, most important, which experimental confirmation can the authors provide able to support all and/or part of their findings? 

Answer 4: We thank the reviewer for this valuable comment. Our study is entirely theoretical and mainly describes the mechanism of action, which is difficult to determine experimentally. However, some experiments such as HOO radical scavenging and XO inhibitory activity assays may be useful to support our findings.
In addition, when analyzing each paragraph of the manuscript, in this reviewer’s opinion:
a)    The Introduction presents many too generic, although interesting, considerations. Instead, the authors must focus principally on the matter of the study both from a theoretical and experimental point of view. The above reported information must be added, together with something about the adopted techniques. Finally, the most important results must be reported at the end of the paragraph, possibly to capture the reader’s interest.

Answer: We thank the reviewer for this constructive suggestion. The introduction has been adapted in line with the reviewer 's recommendations.

b)    Materials and Methods are presented in a quite confusing way. This reviewer suggests, for more clarity, to separate the Methods, that is theory and equations, from Materials. In particular the authors never report the ‘ materials’ used to perform the experiments but sometimes only the software adopted. A manuscript must absolutely reported what the authors used to perform experiments. 

Answer: We thank the reviewer for this constructive suggestion. As our study does not report any experiments, the "Materials and methods" section has been named " Computational details".

c)    Results and Discussion. As underlined above, the comparison between literature data and the present study doesn’t clearly appear, so that the novelty of the study is not reported. Sometimes references must be added.

Answer: We thank the reviewer for this constructive suggestion. Some comparisons with the literature have been added to the revised manuscript

The Figures are not described. Each Figure must be self consistent: the reader must understand from the Capture the meaning of the Figure and at the same time colors, symbols etc must be clearly described. 

Answer: We thank the reviewer for this constructive suggestion. Each figure has been described in the revised manuscript.

d)    The authors report (see Abstract) 5-CQA carrying catechol moiety was found to be significantly more potent than 5-FQA in both physiological environments. The significance level is not reported and a Statistics paragraph lack in the manuscript. So, which kind of significance are the authors referring to? 

Answer: We would like to thank the reviewer for this valuable comment. As we did not conduct a statistical study and the level of significance is unclear, we have omitted the term "significantly" in the revised manuscript.

e) Conclusions. Due to the complexity of the study and the multiplicity of data and methods, this paragraph is too short. A clear and exhaustive synthesis of the study and the methos is strongly suggested before the concluding remarks. At the same time an indication of further research in the field is suggested too. 

Answer: We would like to thank the reviewer for this valuable comment. We appreciate your comments about the complexity of the study and the need for a more comprehensive overview of the data and methods. We have therefore addressed this concern by expanding the conclusions section to provide a clearer and more comprehensive overview of our study and the methodologies employed. In addition, we have included a section on future research directions to indicate potential areas for further exploration in this area.

Reviewer 3 Report

1. Bond dissociation enthalpy (BDE), one factor that best describes the hydrogen atom transfer mechanism of any antioxidant, is absent from this publication. Thus, talk about this parameter.

2. The proton affinity value is described by the authors, although they do not explain it well. Authors must explain how PA was determined and how they characterize the antioxidant activity. 

3. ETE (electron transfer enthalpy) is crucial when describing the electron transfer-mediated mechanism. ETE data, however, was absent. So, writers ought to address this. 

4. The authors should describe all parameters and the influence of each parameter in Tables 1 and 2, as this information is absent in the current version of the publication.  

5. A footnote in the table should explain each used acronym. 

6. There is no HOMO orbital presentation. The qualitative information of the active site for the effects of radical scavenging is described by the HOMO energy calculation. 

7. RMSF, Hydrogen bonds, RG, and SASA with RMSD are only a few interaction parameters that explain the molecular dynamics of ligand and protein interaction. Discuss this issue scientifically, then. 

8. This work does not compare the various MD parameters between apo-protein and protein-ligand interaction, which is necessary to describe the MD. To fully comprehend, read the following article: https://doi.org/10.1016/j.compbiomed.2022.105468  

Moderate English correction is needed. 

Author Response

Point 1. Bond dissociation enthalpy (BDE), one factor that best describes the hydrogen atom transfer mechanism of any antioxidant, is absent from this publication. Thus, talk about this parameter. 2. The proton affinity value is described by the authors, although they do not explain it well. Authors must explain how PA was determined and how they characterize the antioxidant activity.  3. ETE (electron transfer enthalpy) is crucial when describing the electron transfer-mediated mechanism. ETE data, however, was absent. So, writers ought to address this. 

Answer: We thank the reviewer for these valuable comments. BDE, PA, ETE, IP and PDE are well-known intrinsic indicators that describe the thermodynamic preference of specific anti-radical mechanisms such as HAT, SETPT and SPLET. These parameters are characteristic of the antioxidant and also depend on the environment. Although these parameters can provide valuable information on the preferred mechanism, they are limited because they do not take into account the reactivity of the radical species, which may be the main factor determining the antiradical mechanism. Furthermore, these parameters are widely described in the literature for such phenolic compounds. Thus, in our study, we focused solely on the reactivity towards the hydroperoxyl radical using both thermodynamic and kinetic studies. This approach is more accurate than that based on intrinsic reactivity indicators and is widely used by our team and other researchers specialized in this field.

Point 4. The authors should describe all parameters and the influence of each parameter in Tables 1 and 2, as this information is absent in the current version of the publication. 5. A footnote in the table should explain each used acronym.

Answer: We thank the reviewer for this constructive suggestion. The meaning of all the parameters used in Tables 1 and 2 is described in the computation details section. Furthermore, the acronyms used have been explained in the footnotes and legends of the Tables.

Point 6. There is no HOMO orbital presentation. The qualitative information of the active site for the effects of radical scavenging is described by the HOMO energy calculation. 

Answer: We thank the reviewer for this valuable comment. As indicated above, HOMO is also an intrinsic indicator of reactivity and is not taken into account in our study. We will explore all these parameters in future studies.

Point 7. RMSF, Hydrogen bonds, RG, and SASA with RMSD are only a few interaction parameters that explain the molecular dynamics of ligand and protein interaction. Discuss this issue scientifically, then. 

Answer: We appreciate the reviewer's comment regarding the explanation of the MD analysis. The RMSF and RMSD metrics are two parameters that allow measuring the stability of the protein and the ligand along the whole trajectory. Also, these measurements combined with the interactions established between the protein and the ligand are used in most of the papers that use Maestro-Desmond. With this information, researchers can analyze in detail the key residues of the protein.

Point 8. This work does not compare the various MD parameters between apo-protein and protein-ligand interaction, which is necessary to describe the MD. To fully comprehend, read the following article: https://doi.org/10.1016/j.compbiomed.2022.105468 

Answer: We thank the reviewer for the comment regarding the analysis of the MD. We have included the MD with the protein without ligand and the analysis of this simulation. 

Round 2

Reviewer 2 Report

I think the manuscript in the submitted revised version suitable for publication in Antioxidants Journal.

minor english editing required

Reviewer 3 Report

The present manuscript is well improved.